# Pivoting Novel Exosome-Based Technologies for the Detection of SARS-CoV-2

**DOI:** 10.3390/v14051083

**Published:** 2022-05-18

**Authors:** Christine Happel, Chariz Peñalber-Johnstone, Danilo A. Tagle

**Affiliations:** National Center for Advancing Translational Sciences, National Institutes of Health, Bethesda, MD 20892, USA; christine.happel@nih.gov (C.H.); chariz.johnstone@nih.gov (C.P.-J.)

**Keywords:** SARS-CoV-2, COVID-19, exosome, extracellular vesicle, diagnostics, technology, biomarker

## Abstract

The National Institutes of Health (NIH) launched the Rapid Acceleration of Diagnostics (RADx) initiative to meet the needs for COVID-19 diagnostic and surveillance testing, and to speed its innovation in the development, commercialization, and implementation of new technologies and approaches. The RADx Radical (RADx-Rad) initiative is one component of the NIH RADx program which focuses on the development of new or non-traditional applications of existing approaches, to enhance their usability, accessibility, and/or accuracy for the detection of SARS-CoV-2. Exosomes are a subpopulation of extracellular vesicles (EVs) 30–140 nm in size, that are critical in cell-to-cell communication. The SARS-CoV-2 virus has similar physical and molecular properties as exosomes. Therefore, the novel tools and technologies that are currently in development for the isolation and detection of exosomes, may prove to be invaluable in screening for SARS-CoV-2 viral infection. Here, we describe how novel exosome-based technologies are being pivoted for the detection of SARS-CoV-2 and/or the diagnosis of COVID-19. Considerations for these technologies as they move toward clinical validation and commercially viable diagnostics is discussed along with their future potential. Ultimately, the technologies in development under the NIH RADx-Rad exosome-based non-traditional technologies toward multi-parametric and integrated approaches for SARS-CoV-2 program represent a significant advancement in diagnostic technology, and, due to a broad focus on the biophysical and biochemical properties of nanoparticles, the technologies have the potential to be further pivoted as tools for future infectious agents.

## 1. Introduction

Severe acute respiratory syndrome coronavirus 2 (SARS-CoV-2) is a novel coronavirus that emerged in 2019 and is the causative agent of the highly transmissible and highly pathogenic coronavirus disease 2019 (COVID-19) [1]. The novel SARS-CoV-2 is a beta-coronavirus that has a ~30 kb positive-sense single-stranded RNA genome which shares ~80% sequence identity with SARS-CoV-1 and ~50% sequence identity with MERS-CoV [2]. The virus encodes four structural proteins—spike (S), envelope (E), membrane (M), and nucleocapsid (N), as well as several accessory proteins [3]. The SARS-CoV-2 virion is approximately 50–140 nm in size [4,5]. The large number of homo-trimeric viral spike proteins (S) located on the surface of the virion facilitate viral entry into target cells [6]. Interaction between the SARS-CoV-2 S protein receptor-binding domain (RBD) with the human ACE2 receptor mediates viral cell entry, with the assistance of cellular serine protease TMPRSS2 [7,8,9]. Once the virus enters the cell, the viral RNA genome is replicated, and structural proteins are synthesized, assembled, packaged, and viral particles are released. SARS-CoV-2 can be found throughout the body in the upper airways, lungs, blood, gastrointestinal tract, oral cavity, among other locations [10,11,12]. SARS-CoV-2, similar to other RNA viruses, continually mutates, and new variants frequently emerge. The emergence of new virus variants (such as Delta (B.1.617.2), and omicron (B.1.1.529 and its subvariant BA.2)) has become a new crucial point of interest in the ongoing COVID-19 pandemic.

As of May 2022, The U.S. Centers for Disease Control and Prevention (CDC) COVID Data Trackers have reported 81 million cases of COVID-19 in the United States, with over 990,000 total deaths [13]. Worldwide, the WHO has reported more than 515 million cases with at least 6 million deaths [14]. United States Food and Drug Administration (FDA)-authorized COVID-19 diagnostic testing is critical for slowing the spread of the SARS-CoV-2. Thus, there is an urgent public health need for increased COVID-19 testing capacity as well as the ability to detect any potential new variants that may escape current testing regimens [15]. To meet these needs, the National Institutes of Health (NIH) supports the development of COVID-19 diagnostics through a multipronged approach with the goal to speed innovation in the development of COVID-19 testing.

## 2. NIH RADx Initiative

In April 2020, the National Institutes of Health (NIH) launched the Rapid Acceleration of Diagnostics (RADx) initiative to speed innovation in the development, commercialization, and implementation of technologies for COVID-19 testing [16]. The RADx program has four components: RADx-ATP (Advanced Technology Platforms), RADx-Tech, RADx-UP (Underserved Populations), and RADx-Rad (Radical). RADx–ATP supports the scale-up of advanced technologies that can achieve immediate and substantial increases in testing capacity through establishment of ultra-high throughput clinical laboratories. RADx-Tech aims to identify and accelerate the development, scale-up, and deployment of innovative point-of-care technologies whether it be improved clinical laboratory tests or innovative point-of-care and home-based tests within a short timeframe. RADx-UP focuses on improving access to testing in underserved and vulnerable populations. Finally, RADx-Rad focuses on accelerating fundamental research to develop novel, non-traditional approaches for COVID-19 testing and surveillance in non-traditional settings, such as homes, public events, airports, and schools.

## 3. RADx-Rad

The goal of RADx-Rad is to address current gaps in COVID-19 testing. There are three main types of COVID-19 diagnostics: nucleic acid, antigen, and antibody tests. Nucleic acid tests (such as polymerase chain reaction (PCR)-based, or next generation sequencing (NGS) based tests) detect viral genetic material and are the laboratory diagnostic method that was the most quickly established during the SARS-CoV-2 pandemic [17]. Real time PCR (RT-PCR) tests are the current gold-standard in COVID-19 diagnostics and are used by the U.S. Food and Drug Administration (FDA) as the benchmark by which all other potential tests should measure up. RT-PCR tests also require skilled personnel for testing and sample collection, expensive special reagent kits, and a costly centralized laboratory infrastructure which can be a limiting factor during a global pandemic, resulting in long turnaround times to get results. Despite the challenges, RT-PCR-based molecular diagnostics currently remains the most accurate and most sensitive solution available for the detection of SARS-CoV-2. 

Direct antigen tests detect surface proteins from the SARS-CoV-2 virus to diagnose an active or acute infection. Antigen tests produce results faster than molecular tests, with fewer laboratory requirements, but the sensitivity for antigen tests is lower than RT-PCR tests and can significantly vary between tests [18,19,20]. Most antigen tests are qualitative by nature (return a yes/no result) and are known to have a high false negative rate [21]. However, antigen tests are more easily used as at home and point-of-care diagnostics, and the rapid diagnosis of COVID-19 is essential to reduce the spread of COVID-19. 

Antibody tests look for the presence of SARS-CoV-2 antibodies in a given sample to determine if an individual has had a past infection with the virus that causes COVID-19. Antibody tests cannot be used to diagnose whether someone currently has COVID-19. Although serology tests cannot confirm the presence of active virus, they can detect prior exposure. Antibody tests also have a longer detection window and antibodies are more stable for sample collection, transport, and storage. Disadvantages of serology tests in general include their high false positive rate, cross-reactivity with similar viruses, and low sensitivity [22,23,24,25,26,27,28]. 

COVID-19 testing limitations have emerged throughout the pandemic and includes supply chain issues, shortages of standard reagents (such as PCR reagents, personal protective equipment (PPE)), as well as the availability of trained personnel, testing capacity limitations of clinical laboratories, and testing limitations with the emergence of viral mutants. Given the gaps in the current state of diagnostic tests for COVID-19, coupled with severe supply chain issues, there is a clear need for the development of reliable, reproducible, and accessible diagnostic methods that are more sensitive for the detection of SARS-CoV-2 and that can also be deployable in anticipation of future pandemics. In August 2020, the RADx-Rad program issued 13 different NIH funding opportunity announcements (FOAs). In total, 49 RADx-Rad awards were made to 61 institutions in December of 2020. Technologies developed under RADx-Rad includes rapid detection devices, home-based testing technologies, and novel surveillance methods with the goal of supporting new, non-traditional approaches and the application of tools to increase and improve COVID-19 testing. The program also supports new or non-traditional applications of existing technologies to make them more user-friendly, accessible, or accurate. This manuscript specifically focuses on a single funding opportunity toward the development of exosome-based non-traditional technologies towards multi-parametric and integrated approaches for the SARS-CoV-2 program (RFA-OD-20-018). 

## 4. Exosomes

Extracellular vesicles (EVs) are a heterogeneous group of cell-derived membranous structures that transfer biological cargo to local and distant recipient cells within the body to facilitate intercellular communication. These biovesicles are released by virtually all cell types and shuttle bioactive proteins, lipids, and nucleic acids (including extracellular RNA) to distant cells. EVs have been found in biofluids such as plasma, urine, semen, saliva, bronchial fluid, cerebral spinal fluid, breast milk, serum, amniotic fluid, synovial fluid, tears, lymph, and bile [29,30,31,32,33,34,35,36,37,38,39,40,41,42]. Extracellular vesicles is a broad term used for particles released from the cell that are encased in a lipid bilayer; however, there are multiple EVs subtypes which can be differentiated based on their size, biogenesis, release pathways, cargo, and function [43]. Exosomes are a specific subpopulation of EVs that are approximately 30–140 nm in size and of endosomal origin [44]. Functionally, exosomes serve an important role as carriers of proteins and nucleic acids in intercellular communication [45]. EVs and exosomes may serve as diagnostic and prognostic biomarkers for a number of different diseases including cancer, cardiovascular disease, and neurodegenerative diseases [46,47,48,49]. In response to viral infection, exosomes can allow the host to produce effective immunity by activating antiviral mechanisms and transporting antiviral factors between adjacent cells [50]. Viruses can also hijack the exosomal pathway to exploit cellular replication mechanisms and further spread infection throughout the body. Exosomes have been implicated in COVID-19 viral infection due to their ability to traffic critical mediators such as ACE2 receptors and TMPRSS2, making recipient cells susceptible to viral infection [51,52,53,54].

## 5. Novel Exosome-Based Technologies

A major challenge in the field of exosomes and EVs is the improvement in EV separation technologies. The heterogeneity of EVs and the large number of EV subpopulations are significant barriers to understanding the contribution of each specific EV subtype in different pathological systems [55]. Different EV subtypes cannot be fully separated according to size or density because of overlapping physical characteristics [56,57]. The separation of exosomes specifically presents multiple technical challenges due to their nanoscale size and a lack of broadly applicable molecular markers [58,59]. Size-based separation methods for exosome isolation (such as ultracentrifugation) often result in the co-elution of other EV subtypes, as well as other nanoparticles. The NIH Common Fund’s Extracellular RNA Communication program seeks to address this through the development of improved technologies to isolate different carriers of extracellular RNA (exRNA), including EVs and EV subpopulations such as exosomes.

In addition to the isolation of EVs, these novel EV isolation technologies have the ability to analyze the bioactive cargo to further understand EV function in intercellular communication. The NIH Common Fund’s Extracellular RNA Communication program and the Innovative Molecular Analysis Technologies (IMAT) program (supported by NCI), aim to accelerate the development of exRNA, EVs, and exosomes as potential therapeutics and diagnostics through the development of novel tools and technologies. While the NIH Common Fund’s Extracellular RNA Communication program is focused on the role of exRNA in intercellular communication, the novel technologies developed under the Extracellular RNA Communication program have the capacity for high-throughput isolation and characterization of carrier-specific genomic, proteomic, and lipidomic signatures leading to the identification of associated cargo that may confer targeting or functionality.

The current SARS-CoV-2 virus has similar physical and chemical properties as exosomes. SARS-CoV-2 virions are of similar size as exosomes and contain a single-stranded RNA viral genome that can be utilized for the detection of SARS-CoV-2 (Figure 1). Since exosomes resemble many viruses both structurally and functionally, the novel tools and technologies, developed under the IMAT program, the Extracellular RNA Communication program, and other similar programs, may prove to be invaluable in screening for SARS-CoV-2 viral infection and COVID-19, as they were designed for the isolation, detection, and analysis of single vesicles and will inherently have the specificity and sensitivity to detect low viral titers.

## 6. RADx-Rax Exosome Technologies

In August 2020, a funding opportunity was released to address the application of exosome-based technologies to SARS-CoV-2 diagnostics (RFA-OD-20-018). The goal of this program was to seek new ways to identify the current SARS-CoV-2 virus, as well as other potential new and emerging viruses, using newly developed technologies for single extracellular vesicle, exosome, and exRNA isolation and analysis. Since many exosome isolation technologies are already currently in development, NIH aimed to leverage previous investments in these technologies. These novel technologies have already established proof-of-concept in the isolation and analysis of exosomes, and, when pivoted to SARS-CoV-2, will have the potential to address gaps in current COVID-19 testing, as well as supply chain issues associated with traditional technologies. The program was focused on diagnostic development with the intention that proposed technologies will utilize non-invasive, point-of-care sample collection from body fluids to develop non-invasive, reliable, and reproducible COVID-19 diagnostic tests. Highly sensitive, rapid, detection technologies have been an unmet need during the COVID-19 pandemic. The incorporation of single virion level of detection and analysis would provide enhanced sensitivity compared with the existing gold-standard of COVID-19 molecular diagnostics.

Overall, the program specifically focused on three research objectives: (1) technology development, (2) clinical testing and validation, and (3) development of regulatory approval plans. Under this program, newly developed technologies and approaches for single exosome and exRNA isolation and analyses were to be deployed for detection of SARS-CoV-2 virus RNA and/or protein, and/or the detection of IgA, IgG, and IgM antibodies against the virus. Technologies were to be tested using data training sets from known (1) COVID-19 positive, symptomatic subjects, (2) COVID-19 positive, asymptomatic subjects, and (3) COVID-19 negative subjects, as determined by an FDA-approved method. The validation of technologies and approaches was to use cohorts of COVID-19 patients and unaffected individuals representing population demographics inclusive of sex/gender, race, age, and ethnicity. All clinical cohorts were to specifically include women (in similar ratios to men), underrepresented populations, and pediatric populations. Finally, a plan for the regulatory approval of technologies, tests, and approaches was to be developed and submitted to the FDA Center for Devices and Radiological Health (CDRH) for clinical use through an Emergency Use Authorization (EUA). The technologies under development will be assessed for limit of detection, inclusivity, cross-reactivity, and additional performance-based metrics as required by the FDA for the development of COVID-19 diagnostic tests [60].

Four awards were made under this program, to pivot technologies developed for single EV, exosome, and exRNA isolation and analysis, and reposition them for the detection of SARS-CoV-2 (Figure 2 and Table 1). The four awardees and the technologies being developed under the RADx-Rad exosome-based technologies towards multi-parametric and integrated approaches for SARS-CoV-2 (RFA-OD-20-018) are summarized in Table 1 and Figure 2. Many of the technologies being developed under this program utilize collection of saliva as a biofluid sample for testing, and all employ microfluidics for viral separation, followed by a variety of different detection schemes. The intended use of these technologies ranges from diagnostics intended for high complexity testing laboratories, to point-of-care devices (POCs). Additional information about the funded projects can be found in Table 1 or via the NIH RePORTER (https://reporter.nih.gov/, accessed on 9 May 2022).

While nasopharyngeal swab (NPS) is the current gold-standard of sample collection for SARS-CoV-2 detection, the use of saliva as a biofluid for the detection of SARS-CoV-2 emerged early in the pandemic due to the ease of collection and high levels of virus found in the oral cavity [61,62,63]. The use of saliva as a sensitive method for the detection of SARS-CoV-2, particularly for asymptomatic and mild infections has been demonstrated by multiple groups [64,65,66]. The majority of the technologies being developed under the RADx-Rad Exosome program utilize saliva as the biofluid of choice, taking advantage of the increased sensitivity associated with this biofluid while filling an important public health need for simple and accessible sample collection. However, one awardee under this program is also exploring the use of stool, plasma, and blood (in addition to saliva) as potential biofluids for the detection of SARS-CoV-2.

The technologies being developed within the RADx-Rad Exosome program all utilize microfluidic-based platforms for viral separation and isolation (Figure 2) [67,68,69]. The isolation step is a critical component of the technologies which were originally developed for single EV isolation and analysis; it is the isolation step that differentiates these technologies from current diagnostics, as this additional step allows the isolation of intact viral particles prior to viral detection. Current RNA-based diagnostics (RT-PCR-based assays) cannot differentiate between intact infectious virus or degraded viral remnants. Therefore, RT-PCR assays demonstrate a prolonged detection of viral RNA in clinical specimens due to viral shedding, that does not necessarily indicate infectious potential of the patient [12]. The technologies developed under the RADx-Rad Exosome program directly address this challenge through an enrichment step for intact viral particles (and/or exosomes) prior to viral detection. It is the viral isolation step that results in increased viral detection sensitivity and specificity found in the technologies funded under this program. Ultimately, this isolation step results in the earlier detection of SARS-CoV-2, and a significantly lower limit of detection as compared to current diagnostics.

The detection of SARS-CoV-2 is where these technologies significantly differ from each other and current diagnostics (Figure 2). For the detection of SARS-CoV-2, the projects funded under the RADx-Rad Exosome program utilize a variety of non-traditional technologies such as electrical probe-based detection, loop-mediated isothermal amplification (LAMP), total internal reflection fluorescence (TIRF) microscopy, and surface enhanced Raman spectroscopy (SERS). These are very different than the traditional PCR-based diagnostics for viral detection. Due to the specificity of these detection technologies, not only do all of the technologies developed under this program have the ability to detect SARS-CoV-2, but they can differentiate between the many SARS-CoV-2 variants. This is in contrast to current FDA-approved COVID-19 molecular diagnostics which are heavily sequence-dependent technologies. The emergence of variants, particularly variants that escape current testing regimens, further complicates the testing landscape. The FDA has issued guidance on evaluating the potential impact of emerging and future viral mutations on current SARS-CoV-2 and COVID-19 diagnostics to mitigate issues that may arise [70]. Their guidance suggests test developers design redundant sites of SARS-CoV-2 detection and routinely monitor emerging variants. Looking forward, all of the detection technologies in development under the RADx-Rad Exosome program can also rapidly be adapted for the detection of emerging SARS-CoV-2 variants as well as future viral pandemics.

Finally, all the funded projects have the ability to detect different combinations of SARS-CoV-2 RNA and protein, exosomal RNA and protein, and host antibodies in the development of SARS-CoV-2 multi-parametric assays (Figure 2). This program specifically targeted multi-parametric and integrated approaches for SARS-CoV-2 diagnostic development. Meaning that the diagnostics in development have the ability to measure more than one parameter. In this case, the tests have utility as a COVID-19 diagnostic test, but they can also provide additional clinically relevant information. This could include information such as the presence of broadly neutralizing antibodies, or prognostic biomarkers for severe COVID-19 or Post-Acute Sequelae of SARS-CoV-2 (PASC). The advantage of this multi-parametric approach is the ability for these technologies to provide useful information across multiple stages of SARS-CoV-2 infection, from early diagnosis to antibody production following infection. While RT-PCR tests have the sensitivity to detect SARS-CoV-2 infection at an early stage, individuals can also have detectable SARS-CoV-2 RNA for up to 3 months after illness onset since PCR tests can detect viral RNA fragments even after an individual is no longer infectious [71,72]. Direct antigen tests lack the sensitivity to be utilized as a diagnostic during early infection and are primarily used to diagnose active infection [73]. Antibody tests assess the presence of SARS-CoV-2 antibodies but cannot be used as a COVID-19 diagnostic [22,23]. The development of multi-parametric assays can provide essential information across multiple stages of infection though the use of a single integrated test. Further, the inherent viral isolation step allows for the quantification of the amount of intact virus in a specimen which may inform clinicians and guide patient care [74,75]. Multi-parametric assays have the ability to streamline testing and inform clinical decision making, allowing for early clinical intervention and a resulting reduction in the number of COVID-19 patients that progress to a more severe disease phenotype. The disadvantage of a multi-parametric approach is that it complicates regulatory authorization, as there is no established pathway for these technologies.

The technologies developed under RADx-Rad are new technologies that need additional development, prior to their commercialization and/or deployment. This is also true for the RADx-Rad exosome-based technologies program. Due to the novel nature of these technologies, many hurdles remain including regulatory, commercialization, scale-up, and manufacturing challenges.

## 7. Regulatory Considerations

One of the first major hurdles for new diagnostics is the regulatory approval process that is overseen by the U.S. Food and Drug Administration (FDA). On 4 February 2020, the Health and Human Services Secretary determined that COVID-19 was a public health emergency and issued a notice authorizing the emergency use of in vitro diagnostics for the detection and/or diagnosis of COVID-19. Emergency Use Authorization (EUA) authority was designed to speed up the availability of newly-developed SARS-CoV-2 tests [76].

COVID-19 diagnostic testing has been highlighted as critical over the course of the COVID-19 pandemic [77]. In the United States, the FDA Center for Devices and Radiological Health (CDHR) regulates in vitro diagnostic (IVD) development. To this end, the FDA has developed and continually updates, guidance documents for test developers to clarify expectations for EUA submissions that are tailored to SARS-CoV-2 molecular diagnostics. (Additional information about COVID-19 IVD EUAs can be found here: https://www.fda.gov/medical-devices/coronavirus-disease-2019-covid-19-emergency-use-authorizations-medical-devices/in-vitro-diagnostics-euas, accessed on 9 May 2022). Following the termination of an EUA authorization, COVID-19 molecular tests that intend to be marketed in the US would need FDA pre-market clearance. A Pre-Market Approval (PMA), or 510(k) is a pre-market submission made to the FDA by developers to demonstrate that the device to be marketed as safe and effective [78]. However, EUA approvals can provide a foundation for future PMA or 510(k) applications, both of which have more rigorous standards as compared to an EUA submission.

Technologies being developed under the RADx-Rad exosome-based technologies program are required to develop detailed regulatory approval plans to be submitted to the FDA utilizing data generated during the program. However, currently there is no FDA precedent for the new technologies being developed under this program, creating a hurdle toward FDA approval as the technologies must establish baseline performance and clearly demonstrate that the technologies are both safe and effective [79,80]. To address this need, the FDA is collaborating with the NIH RADx program to evaluate the new diagnostic technologies. RADx-Rad awardees meet with the FDA regarding their pre-EUA and EUA applications and have the ability to engage early with the FDA as they work toward regulatory approval. The ability to introduce these new technologies and have an early, open engagement with the FDA results in time and resources savings for test developers working to meet the regulatory requirements of the FDA. It also allows the FDA to provide feedback on regulatory needs and instills a higher level of confidence in the performance of these new technologies. This is particularly important for novel technologies that may not have an FDA-approved direct comparator. Ultimately, early engagement with the FDA helps to streamline the regulatory approval process and get approved diagnostics to market quicker.

## 8. Commercialization Considerations

Intellectual property (IP) ownership, patents, and licensing are other areas of consideration for molecular diagnostic development. IP is a core component of academic–industry partnerships in the biotechnology industry. A comprehensive IP strategy is critical in technology transfer as new diagnostics move toward commercialization. Patents can be utilized by developers to protect diagnostic inventions and provide protection for potential financial investors. Technology licensing then allows companies to partner with developers to buy access to already existing technologies. This is especially important as new technologies move from academic environments to industry for commercialization. Whether partnering with an established commercial entity or establishing a university research-based startup, an industry partner and a well-constructed commercialization plan is needed to take research innovations to market.

RADx-Rad has a diverse portfolio of research projects (Figure 2), and some projects may be ready to move forward with commercialization and deployment in a relatively shorter timeframe, while others may be best suited for future threats as the technology matures. As part of the RADx-Rad Phase II development plan, RADx-Rad aims to support the efforts of those technologies that are poised to move toward commercialization. This will be done through a number of different pathways. One pathway is through partnering with the RADx-Tech program and leveraging its well-established workflow for the commercialization of COVID-19 diagnostic technologies [81]. Another support pathway is through utilization of the NIH Small business Education and Entrepreneurial Development (SEED) office. The NIH SEED office supports the NIH innovator community in their efforts to turn promising scientific discoveries and technologies into healthcare products that improve patient care and enhance health. The NIH SEED office offers support by teaming with experts in business and product development to offer education and mentorship as teams move from academic discovery to commercial products.

## 9. Scale-Up and Manufacturing Considerations

Developing an in vitro diagnostic (IVD) device from concept through product development requires principles from biology, chemistry, physics, and engineering. Therefore, awards funded under the RADx-Rad exosome technologies program were required to involve activities conducted by multi-disciplinary teams of investigators. Factors such as material compatibility, differences between prototype and production methods, and even distribution logistics are all important considerations for IVD development. Scaling-up involves taking a laboratory-developed technology and crafting a safe, reliable, and economical commercial-scale process. Assays and technologies that are developed in the lab often need further efforts when moved to scale-up. This can be partially mitigated by adjusting the technology for a larger scale early in development. The use of materials aligned with regulatory requirements and supply-chain security can also result in significant time and cost savings during the scale-up process.

Following the transition of an IVD to full-scale production, the developer’s emphasis should shift toward manufacturing. Monitoring and controlling the quality of components and assembly is critical. This is where supply chain management becomes important to prevent manufacturing disruptions. Quality management systems (QMS) and careful documentation of production processes according to the International Organization for Standardization (ISO), FDA, and other regulatory partners ensure compliance. RADx-Tech provides additional commercialization resources to support quality management systems, scale-up, and manufacturing [81]. The additional support from the RADx-Tech team is designed to enable success for each project and ultimately, accelerate the market introduction of diagnostic tests for SARS-CoV-2. The NIH SEED office also offers support to innovators by offering education and mentorship through an Innovator Support Team.

The exosome-based technologies for SARS-CoV-2 utilize microfluidics for separation. Microfluidic technologies are often inexpensive, portable, and the disposable nature of the chips makes them suitable for point-of-care devices. The format of this technology can be very compatible with large-scale production, and fabrication and design components have made advances in recent years that aid in manufacturing. Microfluidics use also allows for the integration of multiple functionalities with the potential to detect multiple analytes in one sample enhancing usability. Microfluidic platforms for POC IVDs show promise for many applications and have the potential to significantly change disease diagnosis and pathogen detection [82]. However, RADx-Rad awardees will need to work with commercial partners on scale-up, manufacturing, and QMS in order to move from the bench to a commercially available diagnostic test.

## 10. Post-Acute Sequelae of SARS-CoV-2 Infection (PASC)

While most individuals infected with SARS-CoV-2 recover fully over a period of a few weeks, some individuals continue to experience a number of symptoms long past the time when they have recovered from the initial stages of COVID-19 illness [83]. This is referred to as Post-Acute Sequelae of SARS-CoV-2 infection (PASC) or “long COVID”. While the definition of PASC is still evolving, according to the CDC it generally includes the persistence of symptoms or development of sequelae approximately 4 weeks from the onset of acute symptoms of COVID-19. A recent publication has highlighted the healthcare needs for individuals who experience multi-organ sequelae of COVID-19 beyond the acute phase of infection [83,84]. What we do not know is who is at an increased risk of developing PASC following acute COVID-19 infection. What is responsible for the shift toward PASC? And why? Are there potential biomarkers that could be assessed to predict individuals who are at risk of PASC?

In various diseases, normal EV cargo content changes as diseases initiate, and progress, altering the types of proteins and RNAs that are packaged [85]. EVs have begun to show utility as a diagnostic for COVID-19 and as prognostic biomarkers for disease severity [86,87,88,89,90,91]. Using an untargeted proteomic approach, a group in Italy recently demonstrated that circulating exosomes are modulated during COVID-19 infection and may be involved in disease pathogenesis [86]. Proteomic analysis of the circulating EVs have demonstrated potential biomarkers (fibrinogen, fibronectin, complement C1r subcomponent, and serum amyloid P-component) in COVID-19 patients vs. healthy controls. These proteins were further demonstrated to be differentially regulated in COVID-19 critical vs. non-critical patients [86]. Another recent study demonstrated increased concentrations of EVs in the plasma of hospitalized COVID-19 patients suggesting an increased need for intercellular communication [92]. While additional studies are needed to accurately assess the contribution of EVs in COVID-19 pathogenesis, this research suggests that exosomes could be potential contributors and mitigators of pathogenesis during COVID-19 infection. However, the involvement of exosomes in PASC is still unclear. The analysis of tissue-specific EVs and their associated cargo may shed light on the intercellular signals that are critical for the shift from an acute COVID-19 infection to PASC, and the prognostic biomarkers that may signal that transition. Longitudinal analysis of exosomal cargo could be explored through -omics approaches leading to information about useful biomarkers. Predictive biomarkers for COVID-19 disease severity may show utility for improved patient stratification and assist in early clinical decisions regarding COVID-19 therapeutics.

Since the original application of the technologies being developed under this RADx-Rad program were originally intended to accelerate the development of exRNAs and EVs as potential therapeutics and diagnostics, it is reasonable to fathom that in addition to the detection of SARS-CoV-2, technologies developed under this program may have the ability to also isolate and assess exosomes as a source of prognostic biomarkers. Microfluidic platforms allow for the simultaneous separation of SARS-CoV-2 virions as well as EVs. Detection technologies can assess changes in the expression of nucleic acids originating from virions and EVs. Changes in EV molecular signatures and cargo may imply distinct changes in COVID-19 disease progression as tissues and organs respond to SARS-CoV-2 infection. This is the advantage of the integrated, multi-parametric nature of these technologies. The technologies in development under the RADx-Rad exosome-based technologies program would allow for both SARS-CoV-2 detection and the concurrent detection of prognostic biomarkers that could indicate a predisposition toward the development of severe disease or PASC. Together, this information could inform healthcare strategies, and potentially contribute to the development of therapies for COVID-19.

## 11. Future Pandemics

In June 2021, U.S. Department of Health and Human Services Secretary Xavier Becerra stated, “the pandemic has demonstrated that the U.S. needs transformative technology that is not only available but also widely accessible” [93]. This statement captures the goal of the RADx-Rad program. The current COVID-19 pandemic has highlighted the essential need to accelerate the development and commercialization of technologies and medical products designed to respond to or prevent public health emergencies.

The COVID-19 pandemic has also challenged the usual models for the development of new diagnostics. Due to overwhelming need, new diagnostic development pipelines and support mechanisms have been established in record time. Collaborative efforts have replaced silos. Over the last two years, many smaller companies and academic groups have responded to the COVID-19 pandemic by pivoting their focus and translating their novel concepts into clinically relevant technologies for COVID-19 diagnosis. These new technologies, alternative testing materials, and innovative solutions, have the potential to impart lasting change in diagnostic testing and healthcare.

Diagnostic tests for future pandemics may look very different than those developed for SARS-CoV-2 and the COVID-19 pandemic. Traditional diagnostics often result in binary readouts (such as a yes/no). The multi-parametric nature of the diagnostics in development under the exosome-based non-traditional technologies for SARS-CoV-2 program is challenging this diagnostic paradigm. The potential for a diagnostic test to be combined with an assay for clinically relevant prognostic indicators, resulting in a single integrated test, has enormous potential. Tests that have high multiplex capabilities could provide clinicians with the information required to make appropriate clinical decisions based on additional clinically relevant information beyond a basic diagnostic test.

Beyond the current COVID-19 crisis, it is anticipated that the technologies advanced through RADx-Rad may also be applicable to other, as yet unknown, infectious agents. The technologies in development under the RADx-Rad exosome-based technologies program are disease agnostic and focus on the isolation and analysis of nanoparticles based on specific biophysical and biochemical properties. Therefore, these technologies can not only be adapted for potential new and emerging viruses but may also be relevant for other applications. The technologies are well-suited to pivot toward applications such as liquid biopsies, or toward EVs and lipid nanoparticle-based therapeutic delivery systems. Leveraging the lessons learned during the current pandemic, we can utilize novel models, pipelines, and technologies to spur innovative solutions for future threats. Ultimately, the RADx-Rad exosome-based technologies towards multi-parametric and integrated approaches for SARS-CoV-2 program may lead to new ways to identify the current SARS-CoV-2 virus as well as other potential future viral pandemics.

## Figures and Tables

**Figure 1 viruses-14-01083-f001:**
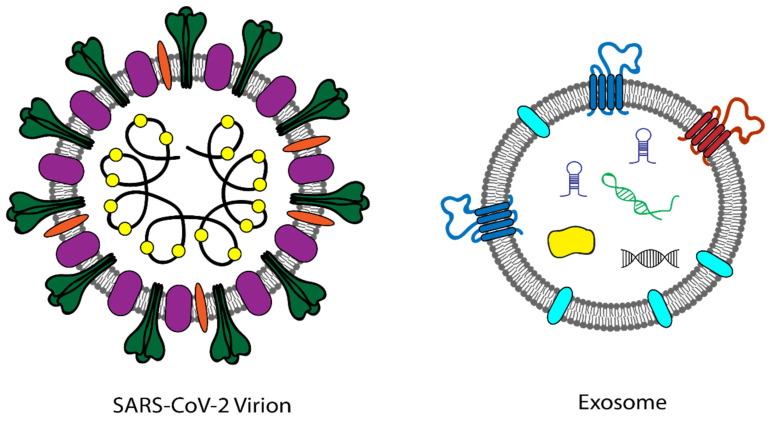
Similar biophysical properties of SARS-CoV-2 and exosomes.

**Figure 2 viruses-14-01083-f002:**
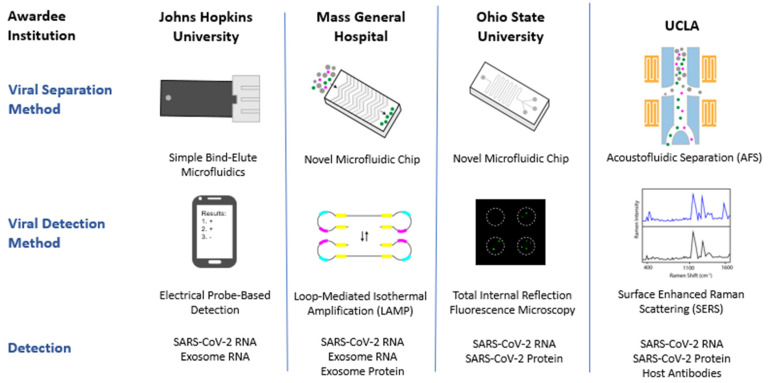
Schematic representing the four projects funded under the RADx-Rad Exosome-based Technologies Towards Multi-Parametric and Integrated Approaches for SARS-CoV-2 FOA. The schematic outlines the awardee institution, the viral separation method, and viral detection method developed under each technology.

**Table 1 viruses-14-01083-t001:** Projects awarded through the RADx-Rad Exosome-based Non-traditional Technologies Towards Multi-Parametric and Integrated Approaches for SARS-CoV-2 program.

RADx-Rad Awardee	Collaborators	Project Title
Samarjit Das— Johns Hopkins University (JHU)	Anubhav Dubey—Sognef Kenneth Witwer—JHU	Exosome-based Non-traditional Technologies Towards Multi-Parametric and Integrated Approaches for SARS-CoV-2
Shannon L. Stott—Massachusetts General Hospital (MGH)	Genevieve Boland—MGH Xu Yu—MGH Seyed Rabi—MGH Jochen Lennerz—MGH	Multi-parametric Integrated Molecular Detection of SARS-CoV-2 from Biofluids by Adapting Single Extracellular Vesicle Characterization Technologies
Eduardo Reategui—Ohio State University (OSU)	L. James Lee—OSU Preeti Pancholi—OSU Shan-Lu Liu—OSU Kai Wang—Institute for Systems Biology Xioakui Mo—OSU	Microfluidics Array Based Sorting, Isolation, and RNA Analysis in Single Extracellular Vesicles
David T. Wong—University of California Los Angeles (UCLA)	Yong Kim—UCLA Ya-Hong Xie—UCLA Samantha Chiang—UCLA David Elashoff—UCLA Jennifer Fulcher—UCLA Wayne Grody—UCLA Feng Li—UCLA Fang Wei—UCLA Otto Yang—UCLA Tony Jun Huang—Duke University	AFS/SERS Saliva-based SARS-CoV-2 Earliest Infection and Antibodies Detection

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
