# Peer review of "Pivoting Novel Exosome-Based Technologies for the Detection of SARS-CoV-2"

_viruses, 2022, doi:10.3390/v14051083_

Round 1

Reviewer 1 Report

The review highlights the potential of exosomes for the detection of SARS-CoV-2. 

Some comments need to be addressed: 

Line 99: “Because antigen tests are qualitative, they can also be inaccurately interpreted”

Why can they be inaccurately interpreted for being qualitative? Maybe those whose results are read visually/by the naked eyed, but there are antigen tests that are read by a reader o that are accompanied by a colour card for avoiding the misinterpretation. Please complete the explanation. 

Line 109: This statement is very general, not all the serology tests give high false positive rate, cross-reactivity with similar viruses and low sensitivity. Serology tests have disadvantages, but not those explained. Or, at least, that statements only apply to some tests, but it is very test-dependent.

Lines 203-205: Three panels of samples are defined to be tested. However, cross-testing with other viruses is missing. Is this panel included in the study?

Line 254: SERS, surface enhanced raman spectroscopy. Please be consistent with the name also indicated in Figure 2.

Line 336: A word is missing between “and” and “are”.

Line 439: Please rephrase the sentence “elucidate indicate prognostic biomarkers”.

Figure 2: In the last column, there is a typing mistake “Raman” instead of “Ramen”. Furthermore, align with correction in line 254.

Are these three technologies already published? Or already applied to any other pathogen?

Reviewer 2 Report

Pivoting Novel Exosome-based Technologies for the Detection 2 of SARS-CoV-2

The review discussed an up to date topic that is interesting to millions of scientists because of COVID-19 pandemic.

  1. Abstract is too long and has repetitive sentences that can be omitted.
  2. Post-Acute Sequelae of SARS-CoV-2 infection (PASC): What is the reason for this part? Authors had to describe the rational of using Exosomes in managing post COVID-19 in more details otherwise this part is not relevant to the main topic.
